# Thermodynamic Assessment of Triclocarban Dissolution Process in *N*-Methyl-2-pyrrolidone + Water Cosolvent Mixtures

**DOI:** 10.3390/molecules28207216

**Published:** 2023-10-22

**Authors:** Diego Ivan Caviedes-Rubio, Claudia Patricia Ortiz, Fleming Martinez, Daniel Ricardo Delgado

**Affiliations:** 1Programa de Ingeniería Civil, Grupo de Investigación de Ingenierías UCC-Neiva, Facultad de Ingeniería, Universidad Cooperativa de Colombia, Sede Neiva, Calle 11 No. 1-51, Neiva 410001, Colombia; diego.caviedesr@campusucc.edu.co; 2Programa de Administración en Seguridad y Salud en el Trabajo, Grupo de Investigación en Seguridad y Salud en el Trabajo, Corporación Universitaria Minuto de Dios-UNIMINUTO, Neiva 410001, Colombia; claudia.ortiz.de@uniminuto.edu.co; 3Grupo de Investigaciones Farmacéutico-Fisicoquímicas, Departamento de Farmacia, Facultad de Ciencias, Universidad Nacional de Colombia, Sede Bogotá, Carrera 30 No. 45-03, Bogotá 110321, Colombia; fmartinezr@unal.edu.co

**Keywords:** triclocarban, solubility, cosolvent, thermodynamics, N-methyl-2-pyrrolidone, water, modeling, simulation

## Abstract

Solubility is one of the most important physicochemical properties due to its involvement in physiological (bioavailability), industrial (design) and environmental (biotoxicity) processes, and in this regard, cosolvency is one of the best strategies to increase the solubility of poorly soluble drugs in aqueous systems. Thus, the aim of this research is to thermodynamically evaluate the dissolution process of triclocarban (TCC) in cosolvent mixtures of {*N*-methyl-2-pyrrolidone (NMP) + water (W)} at seven temperatures (288.15, 293.15, 298.15, 303.15, 308.15, 313.15 and 318.15 K). Solubility is determined by UV/vis spectrophotometry using the flask-shaking method. The dissolution process of the TCC is endothermic and strongly dependent on the cosolvent composition, achieving the minimum solubility in pure water and the maximum solubility in NMP. The activity coefficient decreases from pure water to NMP, reaching values less than one, demonstrating the excellent positive cosolvent effect of NMP, which is corroborated by the negative values of the Gibbs energy of transfer. In general terms, the dissolution process is endothermic, and the increase in TCC solubility may be due to the affinity of TCC with NMP, in addition to the water de-structuring capacity of NMP generating a higher number of free water molecules.

## 1. Introduction

Triclocarban (TCC; 3,4,4-trichlorocarbanilide (Figure 1)) is a broad-spectrum antimicrobial agent, commonly used in personal care products, medical supplies, neonatal products, and even in civil infrastructure [1,2,3]. Despite its high effectiveness against Gram (+) and Gram (−) bacteria, TCC is considered a dangerous agent for public health by the Food and Drug Administration (FDA) and The European Commission (EC) because it is a potent endocrine disruptor [4,5].

In addition to being considered a dangerous agent for human health by the FDA and EC, TCC is listed in the NORMAN list [6] as an emerging contaminant of great danger to aquatic ecosystems, due to its recurrent presence in wastewater, sludge, and runoff [7,8,9].

Solubility is one of the most important physicochemical properties, which allows understanding the biopharmaceutical and pharmacokinetic processes of a drug, in addition to being related to design, formulation, preformulation, recrystallization, quantification, and quality evaluation processes [10,11,12]. On the other hand, in relation to solubility, the use of cosolvency is one of the most used techniques in the pharmaceutical industry to improve the solubility of drugs poorly soluble in aqueous systems [11,13,14]. Furthermore, from cosolvency studies, data of great relevance are determined, such as the dielectric requirement [15] and solubility parameter, as well as better understanding of possible molecular interactions through thermodynamic analysis [16,17] and preferential solvation [18,19,20,21,22].

In addition to interests in industrial processes, solubility also has relevance in environmental processes [23] since, from solubility, some important biological parameters can be determined, such as bioaccumulation [24].

Although TCC poses a risk to human health, some studies open the possibility of its use in the treatment of cancer and HIV [25,26], so the generation of physicochemical information regarding this drug is of great importance. Regarding the use of *N*-methyl-2-pyrrolidone (NMP) as a solvent, this solvent is used in the pharmaceutical industry, in the formulation of drugs for oral and transdermal administration, due to its great solubilizing power, stability, and miscibility with water in all proportions [27,28]. It is also used in extraction, purification, and crystallization processes of drugs [29,30]. Some interesting properties of NMP that make it an eco-friendly solvent are the possibility of being recycled by distillation and water extraction [29,31], in addition to being biodegradable [32] and biosynthesizable [33].

While some solubility data of TCC are reported in the literature, determined in pure solvents [34,35,36], cosolvent mixtures [37,38,39], aqueous systems [40] and in some solubilizing systems [41,42], the physicochemical information on the dissolution process of TCC is not complete, so the study of TCC solubility in {NMP (1) + W (2)} cosolvent mixtures will generate important information on issues of the solubility, cosolvency, and preferential solvation of TCC.

Therefore, the objective of the research is to evaluate the solubility of TCC in different {NMP (1) + W (2)} cosolvent mixtures at different temperatures, which is determined experimentally by UV/Vis spectrophotometry, and from the solubility data, the thermodynamic functions of solutions are calculated using the van ’t Hoff and Gibbs equation. Some of the most relevant results are that maximum solubility is achieved in a cosolvent mixture and that TCC is preferentially solvated by NMP in most cosolvent systems.

## 2. Results

### 2.1. Experimental Solubility ($x_{3}$)

The experimental solubility data of TCC in cosolvent mixtures {NMP (1) + W (2)} are presented in Table 1 and Figure 2, where a strong dependence on the cosolvent composition can be observed. Thus, the minimum solubility is reached in pure water at 288.15 K and the maximum in NMP at 318.15 K.

When analyzing the solubility behavior as a function of temperature, the solubility increases with the increase in temperature, indicating that the TCC solution process is endothermic. Regarding the solubility behavior of TCC as a function of the cosolvent composition, it increases as the solubility parameter of the mixture decreases by adding NMP ($\delta_{1}$ = 23.7 MPa${}^{1/2}$) [43]. Usually, the maximum solubility is reached when the solubility parameters of the drug and the solvent are equal; thus, according to the Fedors group contribution method, TCC has a solubility parameter of 26.5 MPa${}^{1/2}$ [36], so the maximum solubility of TCC should have been reached in a cosolvent mixture and not in pure NMP. However, Delgado et al. determined the TCC solubility parameter experimentally by studying the solubility of TCC in cosolvent mixtures {1,4-dioxane (1) + water (2)}, obtaining a solubility parameter for TCC of 21.92 MPa${}^{1/2}$ [44], so it is conjecturable that the maximum solubility of TCC in cosolvent mixtures {NMP (1) + W (2)} is reached in pure NMP since the solubility parameter of this solvent is greater than 21.92 MPa${}^{1/2}$.

**Table 1 molecules-28-07216-t001:** Experimental solubility of TCC (3) in {NMP (1) + W (2)} cosolvent mixtures expressed in mole fraction at different temperatures (the values in parentheses are the standard deviations). Experimental pressure *p*: 0.096 MPa ${}^{c}$.

$w_{1}$ ${}^{a}$	Temperature/K ${}^{b}$			
	**288.15**	**293.15**	**298.15**	**303.15**
0.00	1.96 × 10^−9 d^	2.38 × 10^−9 d^	2.85 × 10^−9 d^	3.78 × 10^−9 d^
0.05	4.088 (0.020) × 10${}^{-9}$	5.81 (0.07) × 10${}^{-9}$	7.52 (0.10) × 10${}^{-9}$	9.61 (0.10) × 10${}^{-9}$
0.10	8.71 (0.11) × 10${}^{-9}$	10.56 (0.09) × 10${}^{-9}$	15.48 (0.28) × 10${}^{-9}$	20.68 (0.2) × 10${}^{-9}$
0.15	1.66 (0.009) × 10${}^{-8}$	2.368 (0.013) × 10${}^{-8}$	3.143 (0.017) × 10${}^{-8}$	4.155 (0.042) × 10${}^{-8}$
0.20	3.82 (0.04) × 10${}^{-8}$	5.25 (0.04) × 10${}^{-8}$	7.09 (0.07) × 10${}^{-8}$	9.14 (0.04) × 10${}^{-8}$
0.25	8.5 (0.14) × 10${}^{-8}$	10.12 (0.13) × 10${}^{-8}$	14.751 (0.023) × 10${}^{-8}$	20.68 (0.06) × 10${}^{-8}$
0.30	1.669 (0.016) × 10${}^{-7}$	2.305 (0.026) × 10${}^{-7}$	3.32 (0.04) × 10${}^{-7}$	4.47 (0.04) × 10${}^{-7}$
0.35	4.424 (0.012) × 10${}^{-7}$	6.02 (0.06) × 10${}^{-7}$	7.99 (0.04) × 10${}^{-7}$	10.36 (0.04) × 10${}^{-7}$
0.40	0.996 (0.008) × 10${}^{-6}$	1.245 (0.007) × 10${}^{-6}$	1.783 (0.02) × 10${}^{-6}$	2.375 (0.018) × 10${}^{-6}$
0.45	2.363 (0.016) × 10${}^{-6}$	3.29 (0.04) × 10${}^{-6}$	4.27 (0.04) × 10${}^{-6}$	5.58 (0.04) × 10${}^{-6}$
0.50	5.76 (0.05) × 10${}^{-6}$	7.371 (0.023) × 10${}^{-6}$	9.64 (0.14) × 10${}^{-6}$	12.52 (0.05) × 10${}^{-6}$
0.55	1.258 (0.007) × 10${}^{-5}$	1.505 (0.01) × 10${}^{-5}$	2.205 (0.005) × 10${}^{-5}$	2.86 (0.029) × 10${}^{-5}$
0.60	2.827 (0.031) × 10${}^{-5}$	3.679 (0.032) × 10${}^{-5}$	4.884 (0.015) × 10${}^{-5}$	6.542 (0.035) × 10${}^{-5}$
0.65	6.89 (0.07) × 10${}^{-5}$	8.55 (0.09) × 10${}^{-5}$	10.64 (0.11) × 10${}^{-5}$	14.86 (0.05) × 10${}^{-5}$
0.70	1.459 (0.013) × 10${}^{-4}$	1.94 (0.021) × 10${}^{-4}$	2.628 (0.009) × 10${}^{-4}$	3.369 (0.03) × 10${}^{-4}$
0.75	3.456 (0.024) × 10${}^{-4}$	4.449 (0.033) × 10${}^{-4}$	5.73 (0.014) × 10${}^{-4}$	7.29 (0.04) × 10${}^{-4}$
0.80	7.48 (0.08) × 10${}^{-4}$	9.79 (0.1) × 10${}^{-4}$	12.28 (0.2) × 10${}^{-4}$	15.7 (0.13) × 10${}^{-4}$
0.85	1.631 (0.025) × 10${}^{-3}$	2.143 (0.009) × 10${}^{-3}$	2.728 (0.023) × 10${}^{-3}$	3.48 (0.018) × 10${}^{-3}$
0.90	3.63 (0.03) × 10${}^{-3}$	4.669 (0.03) × 10${}^{-3}$	6.004 (0.029) × 10${}^{-3}$	7.43 (0.13) × 10${}^{-3}$
0.95	8.01 (0.06) × 10${}^{-3}$	10.16 (0.09) × 10${}^{-3}$	13.09 (0.10) × 10${}^{-3}$	16.52 (0.10) × 10${}^{-3}$
1.00	1.741 (0.009) × 10${}^{-2}$	2.396 (0.017) × 10${}^{-2}$	2.777 (0.018) × 10${}^{-2}$	3.473 (0.015) × 10${}^{-2}$
$w_{1}$ ${}^{a}$	**Temperature/K** ${}^{b}$			
	**308.15**	**313.15**	**318.15**	
0.00	5.72 × 10^−9 d^	7.48 × 10^−9 d^	9.28 × 10^−9 d^
0.05	12.39 (0.08) × 10${}^{-9}$	15.83 (0.1) × 10${}^{-9}$	23.42 (0.3) × 10${}^{-9}$	
0.10	25.99 (0.27) × 10${}^{-9}$	33.2 (0.3) × 10${}^{-9}$	47.2 (0.8) × 10${}^{-9}$	
0.15	5.48 (0.05) × 10${}^{-8}$	7.13 (0.07) × 10${}^{-8}$	9.25 (0.11) × 10${}^{-8}$	
0.20	11.7 (0.12) × 10${}^{-8}$	16.87 (0.18) × 10${}^{-8}$	22.19 (0.25) × 10${}^{-8}$	
0.25	26.48 (0.27) × 10${}^{-8}$	36.05 (0.37) × 10${}^{-8}$	46.7 (0.6) × 10${}^{-8}$	
0.30	5.81 (0.08) × 10${}^{-7}$	7.68 (0.1) × 10${}^{-7}$	9.49 (0.07) × 10${}^{-7}$	
0.35	15.35 (0.15) × 10${}^{-7}$	19.92 (0.19) × 10${}^{-7}$	23.3 (0.31) × 10${}^{-7}$	
0.40	3.02 (0.02) × 10${}^{-6}$	3.93 (0.03) × 10${}^{-6}$	5.25 (0.01) × 10${}^{-6}$	
0.45	7.028 (0.016) × 10${}^{-6}$	8.877 (0.02) × 10${}^{-6}$	13.18 (0.1) × 10${}^{-6}$	
0.50	15.83 (0.11) × 10${}^{-6}$	22.96 (0.16) × 10${}^{-6}$	27.5 (0.28) × 10${}^{-6}$	
0.55	3.55 (0.04) × 10${}^{-5}$	4.57 (0.05) × 10${}^{-5}$	5.979 (0.024) × 10${}^{-5}$	
0.60	8.281 (0.025) × 10${}^{-5}$	10.415 (0.032) × 10${}^{-5}$	13.57 (0.2) × 10${}^{-5}$	
0.65	18.82 (0.05) × 10${}^{-5}$	23.63 (0.07) × 10${}^{-5}$	31.47 (0.13) × 10${}^{-5}$	
0.70	4.24 (0.04) × 10${}^{-4}$	5.186 (0.004) × 10${}^{-4}$	6.69 (0.05) × 10${}^{-4}$	
0.75	9.176 (0.031) × 10${}^{-4}$	11.49 (0.04) × 10${}^{-4}$	15.4 (0.16) × 10${}^{-4}$	
0.80	20.05 (0.17) × 10${}^{-4}$	25.07 (0.22) × 10${}^{-4}$	32.31 (0.32) × 10${}^{-4}$	
0.85	4.388 (0.032) × 10${}^{-3}$	5.48 (0.04) × 10${}^{-3}$	6.79 (0.05) × 10${}^{-3}$	
0.90	9.54 (0.06) × 10${}^{-3}$	11.89 (0.08) × 10${}^{-3}$	14.78 (0.11) × 10${}^{-3}$	
0.95	20.33 (0.26) × 10${}^{-3}$	25.32 (0.33) × 10${}^{-3}$	31.84 (0.23) × 10${}^{-3}$	
1.00	4.72 (0.04) × 10${}^{-2}$	5.585 (0.018) × 10${}^{-2}$	6.95 (0.11) × 10${}^{-2}$	

^a^ $w_{1}$ is the mass fraction of NMP (1) in the {NMP (1) + W (2)} mixtures free of TCC (3); ^b^ Standard uncertainty in temperature is u(T) = 0.05 K; ^c^ Standard uncertainty in pressure u(p) = 0.001 MPa; ^d^ Values taken from a reference [44].

Figure 2 shows the great cosolvent power of NMP, increasing the solubility of TCC by seven orders of magnitude from pure water to pure NMP. The low solubility of TCC in water may be due to the structuring of water around the non-polar groups of TCC [45]. When adding NMP, the solubility of TCC increases possibly due to two mechanisms. The first is the cosolvent effect of NMP, which in mixtures rich in water weakens the water structure, improving the solubility of TCC [46,47]; this effect is similar to the cosolvent action of ethanol, which is an excellent disruptor of the water structure. The second mechanism, which can occur in mixtures rich in NMD, is the possible formation of an NMP-TCC complex due to hydrophobic interactions [27,46,48] between non-polar groups of both NMP and TCC; NMD presents a relatively large and almost flat sector, which could enhance the formation of this possible NMP-TCC complex, which would theoretically favor the solubility of TCC.

A factor that can intervene in the change in solubility of a drug is the polymorphic changes or formation of solvates [49,50,51]; therefore, it is important to evaluate whether the change in the cosolvent composition promotes the formation of polymorphs.

A classic test to evaluate polymorphic changes is differential scanning calorimetry (DSC). In this vein, Figure 3 shows the DSC spectra of TCC from three solid phases in equilibrium with water, $w_{0.5}$ and NMP and the commercial sample. Table 2 presents the temperature and enthalpy of the fusion results for each of the samples analyzed.

According to the temperature and enthalpy of the fusion results of the four samples, there is a relative deviation no greater than 1.2 %, so it is viable to assume that no polymorphic changes have occurred. In addition, the results agree with those reported in the literature and previous studies by the research group (Table 2).

### 2.2. Ideal Solubility and Activity Coefficients

In addition to evaluating deviations from ideality, from the activity coefficient ($\gamma_{3}$), different molecular interactions that can occur in the solution process can also be assessed (solute–solute: $e_{33}$, solute–solvent: $e_{13}$ and solvent–solvent: $e_{11}$) according to the equation proposed by Hildebrand and Wood Hildebrand and Wood (Equation (Equation 1)) [12,54,55]:(1)$$\ln\gamma_{3}=\left(e_{11}+e_{33}-2e_{13}\right)\frac{V_{3}\phi_{1}^{2}}{RT}$$
where $V_{3}$ is the molar volume of the super-cooled liquid solute, and finally, $\phi_{1}$ is the volume fraction of the solvent, *R* is the gas constant, and *T* is the absolute temperature of the solution. As a first approximation, for relatively low solubilities ($x_{3}$), the term $V_{3}\phi_{1}^{2}R^{-1}T^{-1}$ may be considered constant; thus, $\gamma_{3}$ depends mainly on $e_{11}$, $e_{33}$ and $e_{13}$ [56]. The $e_{11}$ and $e_{33}$ terms are unfavorable for solubility, whereas the $e_{13}$ term favors the solution process. Thus, the activity coefficient is calculated as the ratio between the ideal solubility, which depends exclusively on the physicochemical properties of the drug [57] and the experimental solubility (Equation (Equation 2)):(2)$$\gamma_{3}=\frac{\exp\left\{-\frac{\Delta_{f}H}{R}\left(\frac{T_{f}-T}{T_{f}T}\right)+\frac{\Delta C_{p}}{R}\left(\frac{T_{f}-T}{T}\right)-\frac{\Delta Cp}{R}\ln\left(\frac{T_{f}}{T}\right)\right\}}{x_{3}}$$
where *T* and $T_{f}$ are in K, $\Delta_{f}H$ is the enthalpy of fusion (in kJ·mol${}^{-1}$) of the solute, *R* is the gas constant (in kJ·mol${}^{-1}$·K${}^{-1}$), and $\Delta C_{p}$ is the differential heat capacity of fusion (in kJ K${}^{-1}$·mol${}^{-1}$) [57]. Some researchers like Hildebrand and Scott [58], Neau and Flynn [59], Neau et al. [60] and Opperhuizen et al. [61], assume $\Delta C_{p}$ to be the entropy of fusion ($\Delta_{f}S$), which is calculated as $\Delta_{f}H/T_{m}$.

According to the results of $\gamma_{3}$ (Table 3) the experimental solubility data of TCC in cosolvent mixtures {NMP (1) + water (2)} deviate strongly from ideality, reaching values up to 1.6 $\times \;10^{6}$ in pure water at 288.15 K. As the temperature increases from 288.15 to 318.15 K, $\gamma_{3}$ decreases between 1.6 and 1.8 times possibly due to the increase in molecular agitation increasing the likelihood of particle collision, thus increasing the solubility of TCC and thereby decreasing $\gamma_{3}$ [62,63]; when evaluating $\gamma_{3}$ in terms of cosolvent composition, the increase in NMP in the cosolvent mixture produces a drastic decrease in $\gamma_{3}$, up to 8.69 × $10^{6}$ times in relation to the values in pure water, i.e., solute–solute ($e_{33}$) and solvent–solvent ($e_{11}$) interactions are stronger in more polar media and at the lowest study temperatures. As the polarity of the cosolvent system decreases as a result of the addition of NMP, solute–solvent molecular interactions ($e_{13}$) increase, favoring the solubility of TCC reaching values of $\gamma_{3}$ close to one (near-ideal behavior) between $w_{1}=0.75$ and $w_{1}=0.85$ from the mixture $w_{1}=0.85$, and the values of $\gamma_{3}$ are less than one, indicating a behavior that exceeds ideality, demonstrating the excellent cosolvent power of NMP.

### 2.3. Thermodynamic Functions of Solution

From the experimental solubility data of TCC in cosolvent mixtures {NMP (1) + W (2)}, the enthalpy and Gibbs energy of solution are calculated using the van ’t Hoff–Krug equation (Equations (Equation 3) and (Equation 4)) [64,65,66,67,68]:(3)$$\Delta_{\mathrm{soln}}H{}^{\circ}=-R{\left[\frac{\partial \ln x_{3}}{\partial \left(T^{-1}-T_{\mathrm{hm}}^{-1}\right)}\right]}_{p}$$
(4)$$\Delta_{\mathrm{soln}}G{}^{\circ}=-RT_{\mathrm{hm}}.\mathrm{intercept}$$
where $T_{\mathrm{hm}}$ is the harmonic mean of the study temperatures (302.8 K) (calculated as $T_{hm}=n/\sum_{i=n}^{n}1/T$, where n is the number of temperatures studied), and the intercept is $b=\ln x_{3}^{302.8}$ this value (b) is taken from the linear equation of the modified van ’t Hoff plot ($\ln x_{3}=m\cdot \left(T^{-1}-T_{\mathrm{hm}}^{-1}\right)+b$), where “m” is the slope and “b” is the intercept.

From the values of $\Delta_{\mathrm{soln}}H{}^{\circ}$ and $\Delta_{\mathrm{soln}}G{}^{\circ}$, $\Delta_{\mathrm{soln}}S{}^{\circ}$ is derived from the Gibbs equation as
(5)$$\Delta_{\mathrm{soln}}S{}^{\circ}=\left(\Delta_{\mathrm{soln}}H{}^{\circ}-\Delta_{\mathrm{soln}}G{}^{\circ}\right)T_{\mathrm{hm}}^{-1}$$

From Equations (Equation 5) and (Equation 7), the contribution of the energetic and organizational components to the Gibbs energy is evaluated, and this contribution is corroborated through the Perlovich graphical method [69,70]:(6)$$\zeta_{H}=|\Delta_{\mathrm{soln}}H{}^{\circ}|(|T\Delta_{\mathrm{soln}}S{}^{\circ}|+|\Delta_{\mathrm{soln}}H{}^{\circ}{\left|\right)}^{-1}$$
(7)$$\zeta_{TS}=1-\zeta_{H}$$

According to the results, values greater than 0.5 for $\zeta_{H}$ indicate a greater contribution of the energy component to the solution process, that is, molecular interactions, such as indrogen bridges and van der Waals forces, represent a more relevant role than the effects related to the $T\Delta_{\mathrm{soln}}S{}^{\circ}$ factor.

Table 4 shows the thermodynamic functions of the solution process of TCC (3) in {NMP (1) + W (2)} cosolvent mixtures. The Gibbs energy is positive in all cases and decreases from pure water to pure NMP. It is important to clarify that the standard Gibbs energy of solution is not an indicator of spontaneity; its positive value is a consequence of expressing solubility in molar fraction. Therefore, according to Equation (Equation 4), which technically is $\Delta_{\mathrm{soln}}G{}^{\circ}=-RT_{\mathrm{hm}}\ln x_{3}$, negative values are not obtained in any case. In relation to the uncertainties of the Gibbs energy, which is a propagation of the uncertainties of solubility and temperature, they are relatively small because the uncertainties of the solubility data values are also low.

As for $\Delta_{\mathrm{soln}}H{}^{\circ}$, it is positive in all cases, indicating that the solution process is endothermic. Hence, as the temperature increases, the solubility of TCC increases. In pure water and in water-rich mixtures, $\Delta_{\mathrm{soln}}S{}^{\circ}$ is negative. This may be due to the structuring of water around the non-polar groups of TCC. From $w_{1}=0.15$, $\Delta_{\mathrm{soln}}S{}^{\circ}$ takes positive values, indicating a possible destructuring of water by NMP. Therefore, the enthalpy increases from pure water to $w_{1}=0.25$. This increase indicates the formation of bonds that may be water–water iterations (hydrophobic hydration). From $w_{1}=0.25$, $\Delta_{\mathrm{soln}}H{}^{\circ}$ decreases, possibly due to the destructuring of water.

When evaluating the energetic and organizational contributions to Gibbs energy, the solution enthalpy is the major contributor, especially in water-rich systems, where its influence is greater than 90%. When analyzing the solution process through the Perlovich method (Figure 4), all values are recorded in sectors I ($T\Delta_{\mathrm{soln}}S{}^{\circ}<\Delta_{\mathrm{soln}}H{}^{\circ}$) and VIII ($\Delta_{\mathrm{soln}}H{}^{\circ}>0$ and $T\Delta_{\mathrm{soln}}S{}^{\circ}<0$, $|T\Delta_{\mathrm{soln}}S{}^{\circ}|<|\Delta_{\mathrm{soln}}H{}^{\circ}|$), indicating that enthalpy contributes to the solution process to a greater extent.

### 2.4. Thermodynamic Functions of Transfer

The thermodynamic functions of the hypothetical transfer process of TCC from the medium of higher polarity to the medium of lower polarity (Table 5), are calculated as the difference between the value of the thermodynamic function (*f* is $\Delta_{\mathrm{tr}}G{}^{\circ}$, $\Delta_{\mathrm{tr}}H{}^{\circ}$ or $T\Delta_{\mathrm{tr}}S{}^{\circ}$) of the less polar medium and that of the more polar medium (Equation (Equation 8)):(8)$$\Delta_{\mathrm{tr}}f{}^{\circ}=\Delta_{\mathrm{soln}}f_{\mathrm{less}\;\mathrm{polar}}^{{}^{\circ}}-\Delta_{\mathrm{soln}}f_{\mathrm{more}\;\mathrm{polar}}^{{}^{\circ}}$$

By adding NMP to water, reducing the polarity of the medium, from $w_{1}=0.0$ (pure water) to $w_{1}=0.25$, the Gibbs energy of transfer is negative, indicating the preference of TCC for less polar media. This transfer process is favored by entropy (+) and disfavored by enthalpy (+). From $w_{1}=0.25$ to $w_{1}=1.0$ (pure NMP), the transfer process is also promoted to less polar media ($\Delta_{\mathrm{tr}}G{}^{\circ}$ (−)). In addition, the transfer process is favored by enthalpy (−) and entropy (+). In general terms, the negative value of the Gibbs transfer energy in all cases reflects the positive cosolvent effect (increased solubility) of NMP.

When evaluating the contribution of $\Delta_{\mathrm{tr}}H{}^{\circ}$ and $T\Delta_{\mathrm{tr}}S{}^{\circ}$ to the transfer process by the Perlovich graphical method (Figure 5), except for the process from $w_{1}=0.35$ to $w_{1}=0.40$ (Sector IV: $\Delta_{\mathrm{tr}}H{}^{\circ}<0;\;T\Delta_{\mathrm{tr}}S{}^{\circ}>0;|\Delta_{\mathrm{tr}}H{}^{\circ}|>|T\Delta_{\mathrm{tr}}S{}^{\circ}|$), the transfer process is driven by entropy (Sector II: $\Delta_{\mathrm{tr}}H{}^{\circ}<T\Delta_{\mathrm{tr}}S{}^{\circ}$ and Sector III: $\Delta_{\mathrm{tr}}H{}^{\circ}<0;\;T\Delta_{\mathrm{tr}}S{}^{\circ}>0;\;|T\Delta_{\mathrm{tr}}S{}^{\circ}\left|>\right|\Delta_{\mathrm{tr}}H{}^{\circ}|$). Although in the transfer $\left(w_{1}=0.35\right)\to \left(w_{1}=0.40\right)$ there is double favorability, at this point, the enthalpy of transfer contributes more to the process.

### 2.5. Thermodynamic Functions of Mixing

The solution process involves two sub-processes. The first one consists of the melting of the solute remaining as a supercooled liquid and the formation of the cavity in the solvent to house the solute molecule. The second sub-process consists of the mixing of the liquids, solvent and supercooled solute (Figure 6).

The solution process can be described by Equation (Equation 9):(9)$$\Delta_{\mathrm{Sol}}f{}^{\circ}=\Delta_{\mathrm{mix}}f{}^{\circ}+\Delta_{f}f^{302.8}$$

Clearing $\Delta_{\mathrm{mix}}f{}^{\circ}$ of (Equation 9), we obtain
(10)$$\Delta_{\mathrm{mix}}f{}^{\circ}=\Delta_{\mathrm{soln}}f{}^{\circ}-\Delta_{f}f^{302.8}$$
where *f* represents the Gibbs energy, enthalpy or entropy of mixing, and $f_{f}$ represents the thermodynamic functions of the fusion of TCC (3) and its cooling to the harmonic mean temperature, 302.8 K. As it has been described previously in the literature, in this research, the $\Delta_{\mathrm{soln}}f{}^{\circ}$ values for the ideal solution processes were used instead of $\Delta_{f}f^{302.8}$ [72].

The results of the thermodynamic functions of mixing are tabulated in Table 6. From pure water ($w_{1}=0.00$) to $w_{1}=0.40$, the mixing process discourages the solution process since the values of the enthalpy of mixing correspond to different types of interactions, one of which is the formation of the cavity, which is an endothermic process because energy must be supplied to break the solvent–solvent interactions ($e_{11}$). This process disfavors the solubility of the solute. In pure water, and in mixtures rich in water, the mixing enthapy tends to decrease possibly due to the formation of water–water bonds, as a consequence of hydrophobic solvation around the non-polar groups of the solute, which agrees with the negative values of mixing entropy in aqueous and water-rich mixtures. From $w_{1}=0.40$ to $w_{1}=0.85$, the discouragement of the mixing process to the solution process persists; however, the values of $\Delta_{\mathrm{mix}}G{}^{\circ}$ decrease due to the decrease in the enthalpy of mixing, possibly due to the increase in solute–solvent molecular interactions, which favor the solution process and, unlike the behavior in mixtures rich in water, an entropic favoring (+) is presented. Finally, from $w_{1}=0.90$ to $w_{1}=1.00$, the mixing process favors the solution process ($\Delta_{\mathrm{mix}}G{}^{\circ}$ (−)); in this case, there is enthalpic discouragement and a greater entropic favoring.

According to Perlovich’s analysis, from pure water to $w_{1}=0.85$, the enthalpy of mixing governs the mixing process (Sector VIII: $\Delta_{\mathrm{mix}}H{}^{\circ}>0,\;T\Delta_{\mathrm{mix}}S{}^{\circ}<0,\;|\Delta_{\mathrm{mix}}H{}^{\circ}|>|T\Delta_{\mathrm{mix}}S{}^{\circ}<0|$, Sector I: $\Delta_{\mathrm{mix}}H{}^{\circ}>T\Delta_{\mathrm{mix}}S{}^{\circ}$), and from $w_{1}=0.90$ to $w_{1}=1.00$, the entropy of mixing is the thermodynamic function that drives the mixing process (Sector II: $\Delta_{\mathrm{mix}}H{}^{\circ}<T\Delta_{\mathrm{mix}}S{}^{\circ}$, $|\Delta_{\mathrm{mix}}H{}^{\circ}|<|T\Delta_{\mathrm{mix}}S{}^{\circ}|$) (Figure 7).

### 2.6. Enthalpy–Entropy Compensation Analysis

During the dissolution process, enthalpy changes occur that disfavor the process; however, these enthalpic changes are compensated by entropic changes as a result of non-covalent interactions between the solute and the solvents [73]. Sharp indicated that linear relations between the $\Delta_{\mathrm{soln}}H{}^{\circ}$ and $T\Delta_{\mathrm{soln}}S{}^{\circ}$, usually indicate strongly compensated processes [74].

The compensation between enthalpy and entropy generates defined trends through which the thermodynamic drive of the process can be identified [74,75].

According to Bustamante et al., the enthalpic–entropic compensation can be evaluated by plotting $\Delta_{\mathrm{soln}}H{}^{\circ}$ vs. $\Delta_{\mathrm{soln}}G{}^{\circ}$. Thus, positive slopes indicate that the dissolution process is driven by the enthalpy of solution, and negative slopes indicate an entropic drive [64,76].

In this order of ideas, Figure 8 shows the behavior of the enthalpic–entropic compensation of the NMP solution process in {NMP (1) + W(2)} cosolvent mixtures. From pure water to $w_{1}=0.25$, the process is driven by the solution entropy, and from $w_{1}=0.25$ to pure NMP, the process is driven by the solution enthalpy.

## 3. Materials and Methods

### 3.1. Reagents

In this study, triclocarban (Sigma-Aldrich, Burlington, MA, USA; compound **3**), *N*-methyl-2-pyrrolidone (Sigma-Aldrich, Burlington, MA, USA; the solvent component 1), double-distilled water (component 2) with conductivity lower than 2 μS cm${}^{-1}$, and ethanol (Sigma-Aldrich, Burlington, MA, USA) were used. Table 7 summarizes the sources and purities of the compounds studied.

### 3.2. Preparation of Solvent Mixtures

All cosolvent mixtures {NMP (1) + W (2)} were prepared geometrically (mass fraction) using an analytical balance with a sensitivity of ±0.0001 g (RADWAG AS 220.R2, Krakow, Poland). In amber-colored bottles (capacity 10 mL), 19 mixtures were prepared, varying the mass fraction in 0.05. Three samples were prepared for each mixture.

### 3.3. Solubility Determination

Triclocarban solubility was determined according to the shake-flask method proposed by Higuchi and Connors [77,78,79]; the method is described in detail in some open access publications [80].

In general terms, the samples were saturated by adding an excess of TCC to ensure a liquid phase (saturated solution) and a solid phase (excessive drug). These were then deposited in a recirculation bath (Medingen K-22/T100, Medingen, Germany) at 288.15, 293.15, 298.15, 303.15, 308.15, 313.15 and 318.15 K for 72 h and periodically agitated. Subsequently, an aliquot of each sample was taken by a syringe, filtering the dispersion with a membrane with a pore diameter of 0.45 µm (Millipore Corp. Swinnex-13, Burlington, MA, USA), which was then diluted gravimetrically with ethanol to avoid the precipitation of TCC, followed by quantification by UV/Vis spectrophotometry (UV/Vis EMC-11- UV spectrophotometer, Duisburg, Germany) at 265 nm (wavelength of maximum absorbance) (see Appendix A). Due to the low solubility of TCC in water-rich solvent mixtures ($w_{1}=0.05$ to $w_{1}=0.40$), the concentration was determined by the standard addition method. Thus, 10.00 g ($m_{1}$) of a 10.00 μg/g TCC ($C_{1}$) solution were taken, and 10.00 g of saturated TCC solution ($m_{2}$) (unknown concentration) were added; the mixture ($m_{f}$) was stirred to homogenize, and the absorbance of the final solution ($m_{f}$) was determined. Then, the concentration was calculated using the calibration curve equation ($C_{f}$) (see Appendix A). The concentration of the saturated solution (unknown concentration ($C_{\mathrm{soln-sat}}$)) was determined by means of (Equation (Equation 11)):(11)$$C_{\mathrm{soln-sat}}=\frac{C_{f}\cdot m_{f}-C_{1}m_{1}}{m_{2}}$$

### 3.4. Calorimetric Study

The enthalpy and melting temperature of four TCC samples were determined by differential scanning calorimetry (DSC 204 F1 Phoenix, Munich, Germany). The equipment was calibrated using Indium and Tin as standards, and an empty sealed pan was used as reference. A mass of approximately 10.0 mg of each sample was deposited in an aluminum crucible and placed in the calorimeter under a nitrogen flow of 10 mL min${}^{-1}$. The heating cycle was developed from 323 to 523 K, with a heating ramp of 10 K min${}^{-1}$. The solid samples in equilibrium with the saturated solution were dried at room temperature for 48 h under a continuous stream of dry air.

## 4. Conclusions

The solution process of triclocarban in {NMP (1) + W(2)} cosolvent mixtures is an endothermic process, heavily dependent on the cosolvent composition, demonstrating also the great solubilizing power of NMP, which increases the solubility of TCC up to seven orders of magnitude. This is corroborated by the thermodynamic transfer functions.

The solution process is endothermic, with enthalpic and entropic disfavor in water-rich mixtures and entropic favor in intermediate and NMP-rich mixtures, which can be corroborated with the mixing functions which disfavor the solution process in water-rich and intermediate mixtures and favor it in NMP-rich mixtures.

Finally, the solution process is strongly compensated and is driven by the entropy of the solution in water-rich mixtures, while in intermediate and NMP-rich mixtures, the solution process is driven by the solution enthalpy.

## Figures and Tables

**Figure 1 molecules-28-07216-f001:**
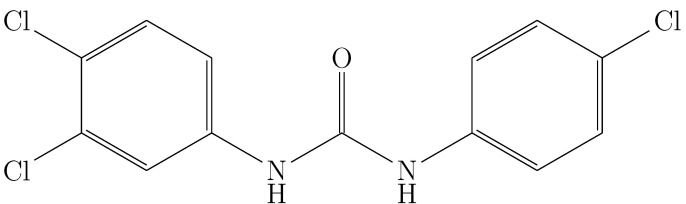
Molecular structure of triclocarban.

**Figure 2 molecules-28-07216-f002:**
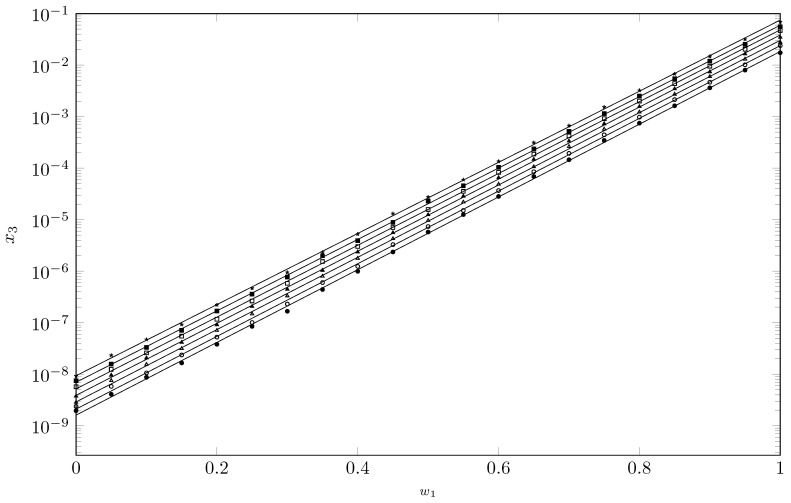
Mole fraction of TCC (10${}^{3}$
$x_{3}$) depending on the mass fraction of NMP in the {NMP (1) + W (2)} mixtures free of TCC. •: 288.15 K; ∘: 293.15 K; △: 298.15 K; ▲: 303.15 K; ◻: 308.15 K; ◼: 313.15 K; 🟉: 318.15 K.

**Figure 3 molecules-28-07216-f003:**
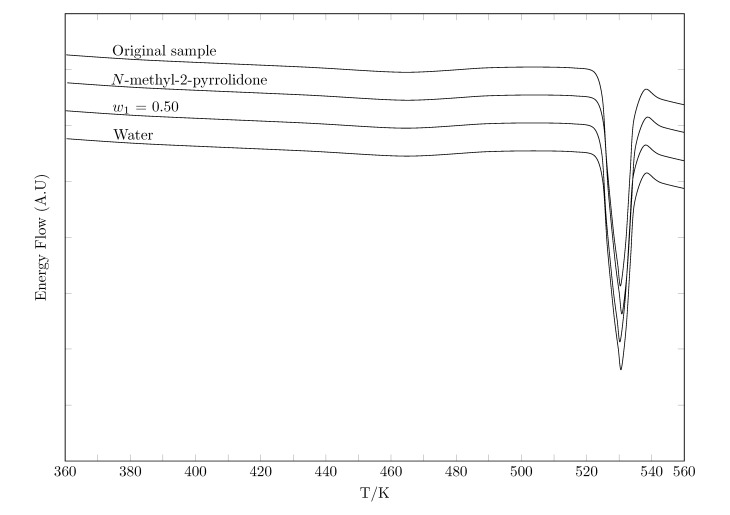
DSC thermograms of TCC.

**Figure 4 molecules-28-07216-f004:**
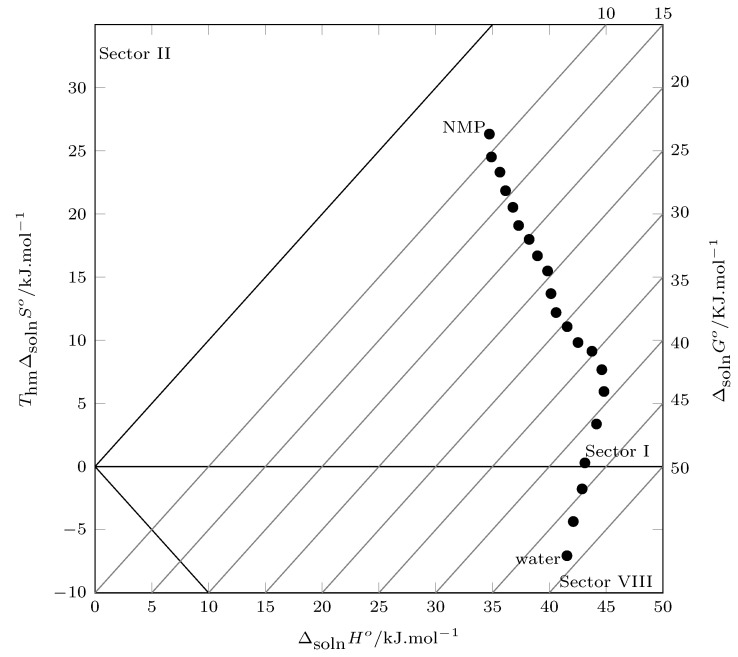
Relation between enthalpy ($\Delta_{\mathrm{soln}}H{}^{\circ}$) and entropy ($T_{\mathrm{hm}}\Delta_{\mathrm{soln}}S{}^{\circ}$) in terms of the process of TCC (3) solution in {NMP (1) + W (2)} cosolvent mixtures at 302.8 K. The isoenergetic curves for $\Delta_{\mathrm{soln}}G{}^{\circ}$ are represented by dotted lines.

**Figure 5 molecules-28-07216-f005:**
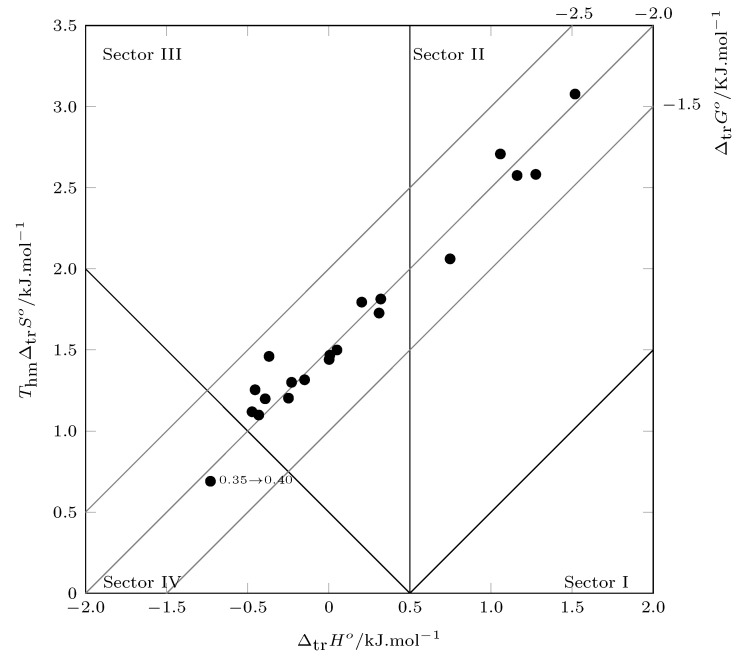
Relation between enthalpy ($\Delta_{\mathrm{tr}}H{}^{\circ}$) and entropy ($T_{\mathrm{hm}}\Delta_{\mathrm{tr}}S{}^{\circ}$) of the process transfer of TCC (3) in {NMP (1) + W (2)} cosolvent mixtures at 302.8 K. The isoenergetic curves for $\Delta_{\mathrm{mix}}G{}^{\circ}$ are represented by dotted lines.

**Figure 6 molecules-28-07216-f006:**
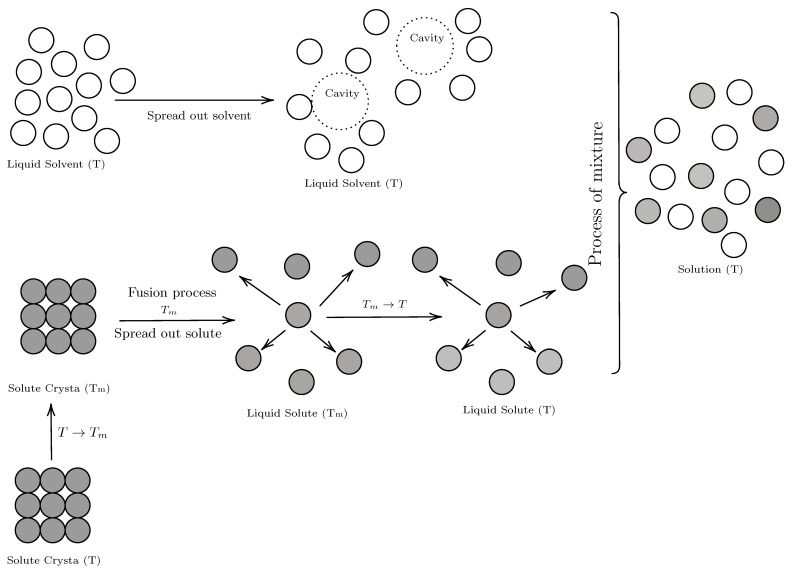
Diagram of the hypothetical of solution process [71].

**Figure 7 molecules-28-07216-f007:**
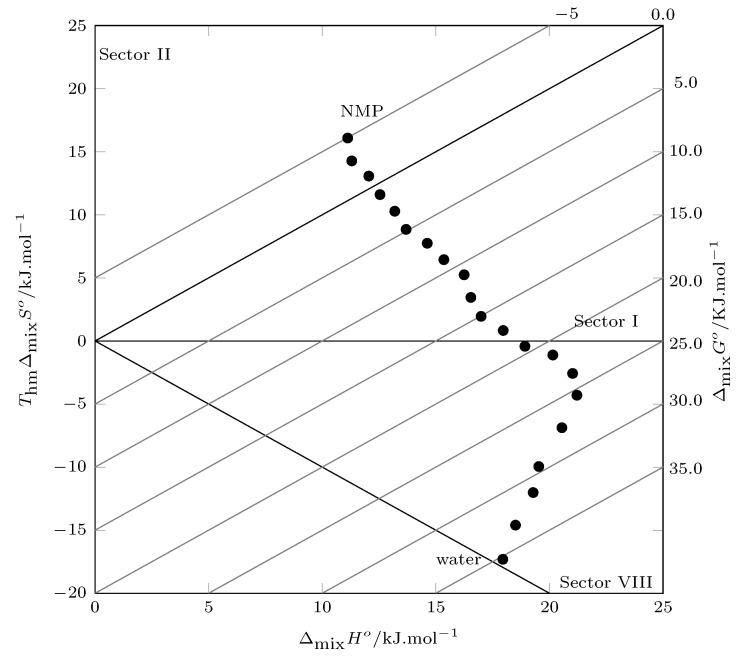
Relation between enthalpy ($\Delta_{\mathrm{mix}}H{}^{\circ}$) and entropy ($T_{\mathrm{hm}}\Delta_{\mathrm{mix}}S{}^{\circ}$) of the process mixing of TCC (3) in {NMP (1) + W(2)} cosolvent mixtures at 302.8 K. The isoenergetic curves for $\Delta_{\mathrm{mix}}G{}^{\circ}$ are represented by dotted lines.

**Figure 8 molecules-28-07216-f008:**
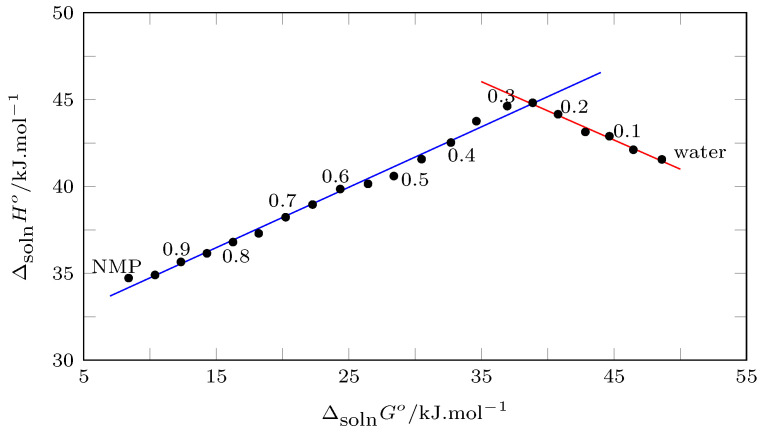
Enthalpy–entropy compensation plot for the solubility of TCC (3) in {NMP (1) + W (2)} mixtures at $T_{\mathrm{hm}}$ = 302.8 K.

**Table 2 molecules-28-07216-t002:** The thermophysical properties of TCC obtained by the DSC.

Sample	Enthalpy of Fusion, $\Delta_{\mathrm{fus}}H$/kJ·mol${}^{-1}$	Melting Point, $T_{\mathrm{fus}}$/K	Ref.
Original sample	41.3 ± 0.5	528.4 ± 0.5	This work
	41.94	528.2	[52]
		527.8	[39]
		525	[44]
		528.15–529.15	[53]
Water	41.9 ± 0.5	527.5 ± 0.5	This work
$w_{0.50}$	42.2 ± 0.5	527.9 ± 0.5	This work
NMP	41.2 ± 0.5	528.6 ± 0.5	This work

**Table 3 molecules-28-07216-t003:** Activity coefficient of TCC (3) in {NMP (1) + water (2)} cosolvent mixtures at different temperatures and pressure *p* = 0.096 MPa.

$w_{1}$ ${}^{a}$	Temperature/K						
	**288.15**	**293.15**	**298.15**	**303.15**	**308.15**	**313.15**	**318.15**
0.00	1.56 × 10${}^{6}$	1.52 × 10${}^{6}$	1.49 × 10${}^{6}$	1.32 × 10${}^{6}$	1.02 × 10${}^{6}$	9.08 × 10${}^{5}$	8.51 × 10${}^{5}$
0.05	7.50 × 10${}^{5}$	6.22 × 10${}^{5}$	5.65 × 10${}^{5}$	5.18 × 10${}^{5}$	4.70 × 10${}^{5}$	4.29 × 10${}^{5}$	3.37 × 10${}^{5}$
0.10	3.52 × 10${}^{5}$	3.42 × 10${}^{5}$	2.74 × 10${}^{5}$	2.41 × 10${}^{5}$	2.24 × 10${}^{5}$	2.05 × 10${}^{5}$	1.67 × 10${}^{5}$
0.15	1.85 × 10${}^{5}$	1.53 × 10${}^{5}$	1.35 × 10${}^{5}$	1.20 × 10${}^{5}$	1.06 × 10${}^{5}$	9.52 × 10${}^{4}$	8.53 × 10${}^{4}$
0.20	8.02 × 10${}^{4}$	6.89 × 10${}^{4}$	5.99 × 10${}^{4}$	5.44 × 10${}^{4}$	4.97 × 10${}^{4}$	4.02 × 10${}^{4}$	3.56 × 10${}^{4}$
0.25	3.61 × 10${}^{4}$	3.57 × 10${}^{4}$	2.88 × 10${}^{4}$	2.41 × 10${}^{4}$	2.20 × 10${}^{4}$	1.88 × 10${}^{4}$	1.69 × 10${}^{4}$
0.30	1.84 × 10${}^{4}$	1.57 × 10${}^{4}$	1.28 × 10${}^{4}$	1.11 × 10${}^{4}$	1.00 × 10${}^{4}$	8.84 × 10${}^{3}$	8.32 × 10${}^{3}$
0.35	6.93 × 10${}^{3}$	6.00 × 10${}^{3}$	5.32 × 10${}^{3}$	4.81 × 10${}^{3}$	3.79 × 10${}^{3}$	3.41 × 10${}^{3}$	3.39 × 10${}^{3}$
0.40	3.08 × 10${}^{3}$	2.90 × 10${}^{3}$	2.38 × 10${}^{3}$	2.10 × 10${}^{3}$	1.92 × 10${}^{3}$	1.73 × 10${}^{3}$	1.51 × 10${}^{3}$
0.45	1.30 × 10${}^{3}$	1.10 × 10${}^{3}$	9.94 × 10${}^{2}$	8.93 × 10${}^{2}$	8.28 × 10${}^{2}$	7.65 × 10${}^{2}$	5.99 × 10${}^{2}$
0.50	5.33 × 10${}^{2}$	4.90 × 10${}^{2}$	4.41 × 10${}^{2}$	3.98 × 10${}^{2}$	3.68 × 10${}^{2}$	2.96 × 10${}^{2}$	2.87 × 10${}^{2}$
0.55	2.44 × 10${}^{2}$	2.40 × 10${}^{2}$	1.93 × 10${}^{2}$	1.74 × 10${}^{2}$	1.64 × 10${}^{2}$	1.49 × 10${}^{2}$	1.32 × 10${}^{2}$
0.60	1.08 × 10${}^{2}$	98.2	87.0	76.1	70.3	65.2	58.2
0.65	44.5	42.3	39.9	33.5	30.9	28.7	25.1
0.70	21.0	18.6	16.2	14.8	13.7	13.1	11.8
0.75	8.87	8.12	7.41	6.83	6.34	5.91	5.13
0.80	4.10	3.69	3.46	3.17	2.90	2.71	2.44
0.85	1.88	1.69	1.56	1.43	1.33	1.24	1.16
0.90	0.845	0.774	0.707	0.67	0.61	0.571	0.534
0.95	0.383	0.356	0.325	0.301	0.286	0.268	0.248
1.00	0.176	0.151	0.153	0.143	0.123	0.122	0.114

^a^ $w_{1}$ is the mass fraction of NMP (1) in the {NMP (1) + W (2)} mixtures free of TCC (3).

**Table 4 molecules-28-07216-t004:** Thermodynamic functions of the solution process of TCC (3) in {NMP (1) + W (2)} co-solvent mixtures at 302.18 K and pressure *p* = 0.096 MPa (the values in parentheses are the standard deviations).

$w_{1}$ ${}^{a}$	$\Delta_{\mathrm{soln}}G{}^{\circ}$/kJ·mol${}^{-1}$	$\Delta_{\mathrm{soln}}H{}^{\circ}$/kJ·mol${}^{-1}$	$\Delta_{\mathrm{soln}}S{}^{\circ}$/J·mol${}^{-1}\cdot$K${}^{-1}$	$T\Delta_{\mathrm{soln}}S{}^{\circ}$/kJ/mol${}^{-1}$	$\zeta_{H}$	$\zeta_{\mathrm{TS}}$
0.00	48.6 ${}^{b}$	41.55 ${}^{b}$	−23.33 ${}^{b}$	−7.06 ${}^{b}$	0.85 ${}^{b}$	0.15 ${}^{b}$
0.05	46.5 (0.4)	42.1 (0.8)	−14.37 (0.30)	−4.35 (0.09)	0.91	0.09
0.10	44.7 (0.5)	42.9 (0.8)	−5.85 (0.13)	−1.77 (0.04)	0.96	0.04
0.15	42.8 (0.3)	43.1 (0.3)	0.962 (0.010)	0.291 (0.003)	0.99	0.01
0.20	40.8 (0.4)	44.2 (0.6)	11.13 (0.18)	3.37 (0.05)	0.93	0.07
0.25	38.9 (0.4)	44.8 (0.7)	19.6 (0.4)	5.94 (0.11)	0.88	0.12
0.30	37.0 (0.4)	44.6 (0.5)	25.3 (0.4)	7.67 (0.13)	0.85	0.15
0.35	34.6 (0.3)	43.8 (0.7)	30.2 (0.6)	9.13 (0.17)	0.83	0.17
0.40	32.7 (0.23)	42.5 (0.5)	32.4 (0.4)	9.82 (0.13)	0.81	0.19
0.45	30.49 (0.21)	41.6 (0.8)	36.6 (0.7)	11.08 (0.22)	0.79	0.21
0.50	28.4 (0.22)	40.6 (0.7)	40.3 (0.7)	12.20 (0.22)	0.77	0.23
0.55	26.45 (0.19)	40.1 (0.6)	45.2 (0.8)	13.70 (0.24)	0.75	0.25
0.60	24.36 (0.17)	39.85 (0.25)	51.2 (0.5)	15.49 (0.14)	0.72	0.28
0.65	22.27 (0.14)	39.0 (0.6)	55.1 (0.9)	16.69 (0.28)	0.7	0.3
0.70	20.24 (0.15)	38.2 (0.4)	59.4 (0.7)	17.99 (0.22)	0.68	0.32
0.75	18.2 (0.1)	37.3 (0.4)	63.0 (0.7)	19.09 (0.22)	0.66	0.34
0.80	16.27 (0.17)	36.8 (0.27)	67.8 (0.9)	20.53 (0.26)	0.64	0.36
0.85	14.3 (0.11)	36.1 (0.1)	72.1 (0.6)	21.85 (0.19)	0.62	0.38
0.90	12.34 (0.1)	35.7 (0.2)	77.0 (0.8)	23.31 (0.23)	0.6	0.4
0.95	10.39 (0.09)	34.9 (0.2)	81.0 (0.9)	24.52 (0.26)	0.59	0.41
1.00	8.39 (0.06)	34.7 (0.6)	87.0 (1.6)	26.3 (0.5)	0.57	0.43
Ideal	13.36 ${}^{b}$	24.05 ${}^{b}$	35.27 ${}^{b}$	10.69 ${}^{b}$	0.692 ${}^{b}$	0.308 ${}^{b}$

^a^ $w_{1}$ is the mass fraction of NMP (1) in the {NMP (1) + W (2)} mixtures free of TCC (3); ^b^ Values taken from a reference [39].

**Table 5 molecules-28-07216-t005:** Thermodynamic functions of transfer of TCC (3) in {NMP (1) + W (2)} cosolvent mixtures at 302.8 K and pressure *p* = 0.096 MPa (the values in parentheses are the standard deviations).

More Polar ($w_{1}$) → Less Polar ($w_{1}$)${\;}^{a}$	$\Delta_{\mathrm{tr}}G{}^{\circ}$/kJ·mol${}^{-1}$	$\Delta_{\mathrm{tr}}H{}^{\circ}$/kJ·mol${}^{-1}$	$\Delta_{\mathrm{tr}}S{}^{\circ}$/kJ·mol${}^{-1}\cdot$K${}^{-1}$	$T\Delta_{\mathrm{tr}}S{}^{\circ}$/kJ·mol${}^{-1}$
0.00 → 0.05	−2.2 (0.4)	0.6 (1.5)	8.9 (0.8)	2.71 (0.24)
0.05 → 0.10	−1.8 (0.7)	0.8 (1.1)	8.53 (0.33)	2.58 (0.10)
0.10 → 0.15	−1.8 (0.6)	0.2 (0.8)	6.81 (0.13)	2.06 (0.04)
0.15 → 0.20	−2.1 (0.5)	1.0 (0.6)	10.16 (0.18)	3.08 (0.05)
0.20 → 0.25	−1.9 (0.5)	0.7 (0.9)	8.5 (0.4)	2.58 (0.13)
0.25 → 0.30	−1.9 (0.6)	−0.2 (0.9)	5.7 (0.6)	1.73 (0.17)
0.30 → 0.35	−2.3 (0.5)	−0.9 (0.9)	4.8 (0.7)	1.46 (0.21)
0.35 → 0.40	−1.9 (0.4)	−1.2 (0.9)	2.3 (0.7)	0.69 (0.21)
0.40 → 0.45	−2.21 (0.32)	−1.0 (0.9)	4.1 (0.8)	1.25 (0.25)
0.45 → 0.50	−2.09 (0.31)	−1.0 (1.0)	3.7 (1.0)	1.12 (0.31)
0.50 → 0.55	−1.95 (0.29)	−0.5 (0.9)	5.0 (1.1)	1.5 (0.32)
0.55 → 0.60	−2.09 (0.25)	−0.3 (0.7)	5.9 (0.9)	1.79 (0.28)
0.60 → 0.65	−2.09 (0.22)	−0.9 (0.7)	4.0 (1.0)	1.2 (0.31)
0.65 → 0.70	−2.03 (0.20)	−0.7 (0.7)	4.3 (1.2)	1.3 (0.4)
0.70 → 0.75	−2.03 (0.18)	−0.9 (0.5)	3.6 (1.0)	1.1 (0.32)
0.75 → 0.80	−1.94 (0.20)	−0.5 (0.5)	4.8 (1.1)	1.4 (0.3)
0.80 → 0.85	−1.97 (0.20)	−0.65 (0.31)	4.3 (1.1)	1.32 (0.32)
0.85 → 0.90	−1.96 (0.15)	−0.49 (0.24)	4.8 (1.0)	1.47 (0.3)
0.90 → 0.95	−1.95 (0.14)	−0.75 (0.27)	4.0 (1.2)	1.2 (0.3)
0.95 → 1.00	−1.99 (0.11)	−0.2 (0.6)	6.0 (1.9)	1.8 (0.6)

^a^ $w_{1}$ is the mass fraction of NMP (1) in the {NMP (1) + W (2)} mixtures free of TCC (3).

**Table 6 molecules-28-07216-t006:** Thermodynamic functions of mixing TCC (3) in {NMP (1) + W (2)} cosolvent mixtures at 302.8 K and pressure *p* = 0.096 MPa (the values in parentheses are the standard deviations).

$w_{1}$ ${}^{a}$	$\Delta_{\mathrm{mix}}G{}^{\circ}$/kJ·mol${}^{-1}$	$\Delta_{\mathrm{mix}}H{}^{\circ}$/kJ·mol${}^{-1}$	$\Delta_{\mathrm{mix}}S{}^{\circ}$/kJ·mol${}^{-1}$K${}^{-1}$	$T\Delta_{\mathrm{mix}}S{}^{\circ}$/kJ·mol${}^{-1}$
0.00	35.25 ${}^{b}$	17.52 ${}^{b}$	−58.55 ${}^{b}$	−17.73 ${}^{b}$
0.05	33.1 (0.4)	18.5 (0.8)	−48.2 (0.3)	−14.6 (0.1)
0.10	31.3 (0.5)	19.3 (0.8)	−39.68 (0.18)	−12.02 (0.06)
0.15	29.5 (0.3)	19.5 (0.3)	−32.87 (0.14)	−9.95 (0.04)
0.20	27.4 (0.4)	20.5 (0.6)	−22.71 (0.22)	−6.88 (0.07)
0.25	25.5 (0.4)	21.2 (0.7)	−14.2 (0.4)	−4.3 (0.12)
0.30	23.6 (0.4)	21 (0.5)	−8.5 (0.4)	−2.57 (0.13)
0.35	21.26 (0.27)	20.2 (0.7)	−3.7 (0.6)	−1.11 (0.18)
0.40	19.34 (0.24)	18.9 (0.5)	−1.4 (0.4)	−0.42 (0.14)
0.45	17.13 (0.22)	18 (0.8)	2.7 (0.7)	0.83 (0.22)
0.50	15.04 (0.23)	17.0 (0.7)	6.4 (0.7)	1.95 (0.22)
0.55	13.09 (0.19)	16.5 (0.6)	11.4 (0.8)	3.45 (0.24)
0.60	11.00 (0.18)	16.24 (0.25)	17.3 (0.5)	5.25 (0.15)
0.65	8.91 (0.15)	15.4 (0.6)	21.3 (0.9)	6.44 (0.28)
0.70	6.88 (0.16)	14.6 (0.4)	25.6 (0.8)	7.74 (0.23)
0.75	4.85 (0.12)	13.7 (0.4)	29.2 (0.7)	8.84 (0.23)
0.80	2.91 (0.18)	13.19 (0.27)	34.0 (0.9)	10.28 (0.26)
0.85	0.94 (0.12)	12.54 (0.15)	38.3 (0.7)	11.6 (0.2)
0.90	−1.02 (0.12)	12.05 (0.19)	43.2 (0.8)	13.07 (0.24)
0.95	−2.97 (0.11)	11.3 (0.19)	47.1 (0.9)	14.27 (0.26)
1.00	−4.97 (0.08)	11.1 (0.6)	53.1 (1.6)	16.1 (0.5)

^a^ $w_{1}$ is the mass fraction of NMP (1) in the {NMP (1) + W (2)} mixtures free of TCC (3); ^b^ Values taken from a reference [39].

**Table 7 molecules-28-07216-t007:** Source and purities of the compounds used in this research.

Chemical Name	CAS ${}^{a}$	Source	Purity in Mass Fraction	Analytic Technique ${}^{b}$
Triclocarban	101-20-2	Sigma-Aldrich	>0.990	HPLC
*N*-methyl-2-pyrrolidone	872-50-4	Sigma-Aldrich	0.998	GC
Water	25322-68-3			
Ethanol	64-17-5	Sigma-Aldrich	0.998	GC

${}^{a}$ Chemical Abstracts Service Registry Number. ${}^{b}$ HPLC is high-performance liquid chromatography; GC is gas chromatography.

## Data Availability

Data are contained within the article or Appendix A.

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
