# Peer review of "Thermodynamic Assessment of Triclocarban Dissolution Process in N-Methyl-2-pyrrolidone + Water Cosolvent Mixtures"

_molecules, 2023, doi:10.3390/molecules28207216_

Round 1

Reviewer 1 Report

In my opinion, several points in the manuscript need revision, as detailed below:

 Major points

a) The interpretation given for the increase of solubility as NMP content increases, in pages 4 and 5, lines 112 to 119 (considering weakening of water structure or some association NMP-TCC due to hydrophobic effect) may be considered for high water content, but not for low water content.

b) The authors are aware of the possible change in polymorphic form (or solvate formation) mediated by the solvent (page 5, lines 120-122). They show DSC heating thermograms obtained for the solid samples remaining after equilibrium (Figure 3). These samples, when collected, are certainly saturated with the solvent. So, my question is: how were the samples treated before obtaining the DSC curves ? Any treatment may result in polymorphic changes (even just letting the sample stay at ambient conditions). X-ray powder diffraction is mandatory to evaluate the solid forms after equilibrium.

c) The authors present uncertainties for pressure and temperature (as footnotes in Table 2), but no uncertainty values are given for the experimental solubility.

d) Discussion in page 8, lines 180 to 188, is quite confuse due to incorrect equations. Here and for the functions of transfer discussion, it is not straightforward the classification in sectors compared to the works by Perlovich cited in the manuscript, references 70 and 71.

e) Considering the mixing properties, a stepwise process is proposed to interpret the solution process. This involves a step that corresponds to the formation of a cavity in the solvent. This often gives a considerable contribution. And certainly its value should be different in water, a quite structured solvent and in an organic solvent. The authors ignore this contribution in their further studies and do not give any justification.

f) Equation (10) does not give the mixing property at temperature T. Terms that depend on the liquid heat capacity at constant pressure are missing.

g) The authors say in page 8, lines 184-185 that the solution process is enthalpy driven. In page 13, lines 234 to 238 they reach different conclusions.

Minor points:

i) In Figure 1 caption, Isoniazid is mentioned instead of triclocarban.

ii) In page 2, line 64 the units of electrical conductivity are incorrect.

iii) In the calorimetric study, calibration details should be included.

iv) Superscript c appears in the heading of Table 2, with two footnote descriptions. The footnote saying that values are taken from reference 50, must be removed.

v) Page 5, line 124 and Figure 3 caption, page 6 – DSC thermograms and not DSC spectra.

vi) Page 6, -reference is missing for equation (1), as well as the identification of several symbols in this equation.

vii) In equation (4), the intercept should be identified.

viii) Equations (5) and (6) are incorrect.

ix) In equation (8), F should be replaced by f.

x) Temperature is missing in Table 5. Why are temperatures in Tables 6 and 7 different ?

Author Response

Below we present the changes suggested by reviewer 1 to the manuscript Thermodynamic assessment of triclocarban dissolution process in N-methyl-2-pyrrolidone + water cosolvent mixtures.
The authors, we appreciate the valuable contribution of the reviewer to our manuscript. Your contributions have improved the quality of the document.

The answers to each of the suggestions can be found in the attached file.

Changes in the manuscript are presented in red for better verification.

Reviewer 2 Report

This is a very accurate and esaustive paper about a specific argument. 

All graphics are of high graphical quality and easily readable.  Perhaps, Table can be formatted better (align numbers to the right!) and/or moved to S.I..

There is a possibility of interference of environmental  water (air humidity) in the preparation of the samples. They are hygroscopic? Please discute this topic. 

Author Response

Thank you very much for your suggestions, they were taken into account.

Reviewer 3 Report

The work presents experimental and theoretical studies on the solubility of TCC in water and NMP mixtures. The solubility of TCC in water and NMP mixtures, including pure solvents, was measured at four different temperatures. Activity coefficients of solute in the solvent mixtures as well as the enthalpy, entropy, and Gibbs energy of solution were calculated from experimental solubility data.

I have the following concerns that should be addressed before the work is suitable for publication:

-          The solubility of TCC in mixtures with w1 ≤ 0.60 is extremely low. What was the calibration curve range? The authors should provide that in the supplementary materials.

-          Which thermodynamic model is Eq.1? The variables should be explained. Besides, the equation is thermodynamically inconsistent.

-          Eq. 3 is not the van’t Hoff equation. The cited reference does not mention a similar equation. Also, it is weird that the logarithm of the experimental solubility data is used. I would expect the equilibrium constant. The authors should provide the source or the derivation of Eq.3.

-          What is the harmonic mean temperature and intercept? How are these calculated/determined?

-          Check Eq. 5. I think the authors mean enthalpy in the first term.

-          The solution thermodynamic properties should be a function of the activities of all components. In Eq. 3-5, the solution thermodynamic properties seem to be a function of TCC mole fraction only. Can the authors explain?

-          What is the motivation of thermodynamic transfer functions analysis? The authors measure the solubility of TCC in a mixture of water and NMP.

-          Eq. 9 and 10 are not explained. What is the second term on the right-hand side, and how was it calculated?

-          The numerical values of heat and Gibbs energy of mixing are extremely large. This further indicates that the formulas used to calculate these thermodynamic properties are incorrect.

Moderate editing of English language required

Author Response

We appreciate the valuable suggestions made by the evaluator. His/her contributions have definitely improved the document.

Each of the observations were taken into account (See attached file)

Reviewer 4 Report

see attached document

Author Response

(The authors gave the same response as above.)

Round 2

Reviewer 3 Report

The authors addressed most of my comments and suggestions. Although I am still skeptical about some of the study findings, in particular, the values of Gibbs energy and heat of mixing, the overall clarity of the manuscript and methods used were improved in the revised manuscript. The manuscript can be published after a few additional minor corrections:

-          Check the value of the intercept in Eq. 5.

-          LN 214: indicating was repeated.

-          The number of figures in Tables 5, 6, and 7 should be checked. Column 5 should be column 3- column 2, but the number of digits differs in these columns.

-          Check the temperature value in the superscript in eq 11

Only minor language editing is needed.

Author Response

  • Check the value of the intercept in Eq. 5. Ans. The discussion on the term "intercept" in equation 5 is expanded. (lines 182-183)
  • LN 214: indicating was repeated. Ans. The repeated word (indicating) is removed. (line 214)
  • The number of figures in Tables 5, 6, and 7 should be checked. Column 5 should be column 3- column 2, but the number of digits differs in these columns. Ans. The number of digits of the uncertainties was determined according to the factor 3-30, taking into account that enough digits should be given so that the last one to the right is largely uncertain (by 3 or more) and the next to the last may be slightly uncertain (by as many as 3); that is, the number of digits given should be such that the limit of error is in the range 3 to 30 in the last two digits. This procedure was taken from: Experiments in Physical Chemistry, Second Edition, Shoemaker, David P.; Garland, Carl W(Page 35)
  • Check the temperature value in the superscript in eq 11 Ans. The superscript was changed from 303.8 to 302.8 (Equation 11)

Reviewer 4 Report

All issues has been addressed properly

Author Response

All issues has been addressed properly

Ans: Thank you very much for your contributions to our manuscript.